# Comparing Person-Specific and Independent Models on Subject-Dependent and Independent Human Activity Recognition Performance [note 1]

**DOI:** 10.3390/s20133647

**Published:** 2020-06-29

**Authors:** Sebastian Scheurer, Salvatore Tedesco, Brendan O’Flynn, Kenneth N. Brown

**Affiliations:** 1Insight Centre for Data Analytics, School of Computer Science and Information Technology, University College Cork, T12 XF62 Cork, Ireland; brendan.oflynn@tyndall.ie; 2Tyndall National Institute, University College Cork, T12 R5CP Cork, Ireland; salvatore.tedesco@tyndall.ie; 3CONNECT Centre for Future Networks and Communications, Tyndall National Institute, University College Cork, T12 R5CP Cork, Ireland

**Keywords:** human activity recognition, machine learning, ensemble methods, boosting, bagging, inertial sensors

## Abstract

The distinction between subject-dependent and subject-independent performance is ubiquitous in the human activity recognition (HAR) literature. We assess whether HAR models really do achieve better subject-dependent performance than subject-independent performance, whether a model trained with data from many users achieves better subject-independent performance than one trained with data from a single person, and whether one trained with data from a single specific target user performs better for that user than one trained with data from many. To those ends, we compare four popular machine learning algorithms’ subject-dependent and subject-independent performances across eight datasets using three different personalisation–generalisation approaches, which we term person-independent models (PIMs), person-specific models (PSMs), and ensembles of PSMs (EPSMs). We further consider three different ways to construct such an ensemble: unweighted, κ-weighted, and baseline-feature-weighted. Our analysis shows that PSMs outperform PIMs by 43.5% in terms of their subject-dependent performances, whereas PIMs outperform PSMs by 55.9% and κ-weighted EPSMs—the best-performing EPSM type—by 16.4% in terms of the subject-independent performance.

## 1. Introduction

This is an extended version of a paper we presented at the 6th international Workshop on Sensor-based Activity Recognition and Interaction (iWOAR’19) [1]. In addition to the experiments, results, and analyses presented in that paper, this paper covers more of the literature in more depth, considers additional ways to combine person-specific models into ensembles, sharpens and deepens the statistical analysis, and expands the discussion by relating our findings to the pertinent literature.

Human activity recognition (HAR) systems are typically evaluated for their ability to generalise to either unknown users (people not represented in the HAR algorithm’s training data) or to known users (people represented in the training data); the former is known as subject-independent and the latter as subject-dependent performance. The subject-independent performance can be estimated by performing a leave-one-subject-out cross-validation across all users in the dataset, and the subject-dependent performance by performing a separate *k*-fold cross-validation for each user. Which performance should be optimised when developing a HAR system depends on how it is going to be commissioned and deployed. If commissioning a HAR system entails obtaining examples of the activities of interest from its end users—the people whose activities are to be recognised by the deployed system—then we should optimise the subject-dependent performance. Intuitively, this suggests that we train a personalised HAR inference model for each user. We refer to models obtained in this manner as person-specific models (PSMs), because they are tuned for a specific person. If, on the other hand, the system is to be deployed without prior commissioning (i.e., without being trained on data from its end users), then it must ship with a HAR model that has been pre-trained on data from a (presumably representative) sample of users. We refer to a model obtained in this manner as a person-independent model (PIM), because its performance is assumed to be independent of the person using it. PIMs are usually evaluated on subject-independent performance (i.e., unknown users), but it is not uncommon to see them evaluated on subject-dependent performance (known users), an approach that corresponds to a scenario wherein it is possible to obtain sample data from (some of) the system’s end users during commissioning, but not possible to identify users (and hence the appropriate PSM) once the system has been deployed. In addition to PIMs and PSMs, we also consider three different types of ensembles of PSMs to assess whether such ensembles can compete with PIMs in terms of subject-independent performance. The advantage of an ensemble of PSMs (EPSM) is that an EPSM can be improved by exploiting data from new users without accessing the dataset that was used to develop the initial model, simply by fitting a PSM for each new user and adding them to the ensemble. This is not the case for a PIM, which requires the dataset used to develop the initial model to exploit data from new users. The distinction between subject-dependent and subject-independent performance is ubiquitous in the HAR literature, and most empirical evaluations of HAR algorithms make it clear which performance measure is being assessed. Intuitively, we hypothesise that methods ought to do better in terms of subject-dependent performance than in terms of subject-independent performance; that PIMs outperform PSMs in subject-independent performance; and that PSMs outperform PIMs in subject-dependent performance. Unfortunately, not many HAR papers report results for more than one combination of the personalisation–generalisation approach (e.g., PIM or PSM), and subject-dependent and subject-independent performance, and none of them report results for all four combinations, making it impossible to verify whether these hypotheses are correct. This paper presents the first empirical comparison of the subject-dependent and subject-independent performances achieved with PIMs and PSMs on multiple (eight) HAR datasets, using four popular machine learning algorithms that have been used extensively and successfully in the HAR literature. The remainder of this paper proceeds as follows. Section 2 summarises the literature pertinent to the topic of subject-dependent and subject-independent performance in HAR, and the state of the art for the benchmark datasets used in this paper. Section 3 describes our methods, including the benchmark datasets, pre-processing, segmentation, feature extraction, and activity inference and evaluation. Section 4 presents the results of our experiments and their analysis. Section 5 discusses our findings in the context of the related works from Section 2, and Section 6 concludes the paper.

## 2. Related Work

In a 2019 survey of 56 papers on deep learning—deep neural, convolutional, and recurrent neural networks, auto-encoders, and restricted Boltzmann machines—for sensor-based human activity recognition, Wang et al. [2] concluded that there is no single “model that outperforms all others in all situations.” Comparing the results from the original studies for three HAR datasets, among them the Opportunity [3] dataset also employed in this paper, their survey identifies four papers [4,5,6,7] as the state of the art. The following paragraph summarises, in chronological order, the results from these four papers with respect to the predictive performance on the HAR datasets also used in this paper. The first of the four papers is the work by Jiang and Yin [4]. The deep convolutional neural network (DCNN) they proposed in 2015, termed DCNN+, recognises human activities from signal and activity images which are obtained by transforming the signals from a single inertial measurement unit (IMU) via the discrete Fourier transform or 2D wavelets. To disambiguate between pairs of classes with confused predictions (i.e., classes with similar large predicted probabilities) they employ binary SVM classifiers. The three considered models (DCNN, DCNN+, and SVM) all achieved a subject-dependent accuracy of >99% on the FUSION dataset [8]. The second paper, published in the same year by Zhang et al. [5], proposes a different DNN for human activity recognition. This DNN recognises human activities from the raw signals acquired from a wearable IMU and the signal magnitude of the accelerometer’s combined three axes. In an empirical comparison, the authors pit their DNN against traditional (i.e., not deep learning) machine learning algorithms with default, untuned hyper-parameters that operate on five statistical features (mean, standard deviation, energy, spectral entropy, and pairwise correlations between the accelerometer axes) extracted from the raw IMU signals. Their results show their DNN achieving a subject-dependent error rate of 18% on the Opportunity dataset, SVM being a close runner-up with an error rate of 19%. The third paper is the work by Ordóñez and Roggen [6], in which they propose a deep convolutional long short-term memory cell (LSTM) model for human activity recognition. Their LSTM outperformed the baseline convolutional neural network (CNN) in terms of subject-dependent performance (F-score: 93% vs. 91%) on the Opportunity dataset. The fourth paper was published in 2016 by Hammerla et al. [7]. In it, the authors compare CNNs, DNNs, and three different types of LSTMs across three benchmark HAR datasets, all of them consisting of data from multiple IMUs per user. Among them are two datasets, the Opportunity and PAMAP2 [9] datasets, which we also use in this paper. This paper is particularly elucidating because of its exploration of how sensitive models are to the various hyper-parameters that determine a deep learning model’s architecture, learning, and regularisation of parameters. The authors explore the hyper-parameter space by randomly sampling hyper-parameter configurations in hundreds or thousands of experiments. The results from these experiments clearly show that deep neural networks are extremely sensitive to hyper-parameter settings, which is illustrated by the differences between each model’s median and best performance. On the Opportunity dataset, the best model’s (LSTM) median score is 17 percentage points lower than its best score, and on the PAMAP2 dataset the best model’s (CNN) median score is seven percentage points lower than its best score. This latter number is the smallest discrepancy between best and median scores across all models and datasets. In their conclusions, the authors concur with Wang et al. [2] in that no single model dominates across all datasets. The best subject-dependent performance on the Opportunity dataset, an F-score of 93%, was achieved with a bi-directional LSTM, whereas the best subject-dependent performance on the PAMAP2 dataset, an F-score of 94%, was achieved with a CNN. In 2019, Jordao et al. [10] evaluated seven state-of-the-art methods published between 2010 and 2016, including the work by Jiang and Yin [4] discussed above, on six publicly available HAR benchmark datasets, including the PAMAP2 and MHEALTH [11] datasets, which are used in this paper. Three of the seven methods entirely rely on deep neural networks, while the remaining four use handpicked features as inputs to classification algorithms. Each method was evaluated with a different data segmentation strategy (overlapping and non-overlapping sliding windows), and its predictive performance estimated via stratified *k*-fold, leave-one-subject-out, and leave-trials-out cross-validation. Leave-trials-out cross-validation, which is discussed in more detail in Section 3, segments data into overlapping sliding windows one trial at a time. A trial corresponds to one individual’s performance of one activity or sequence of activities during the data acquisition. The results show that naïve resampling methods, such as stratified *k*-fold cross-validation, tend to inflate the predictive performance when used with overlapping sliding windows, because the overlap between subsequent windows can appear in both the training and test data. Based on the (unbiased) subject-dependent performance, estimated via leave-trials-out cross-validation, and the subject-independent performance, estimated via leave-one-person-out cross-validation, the authors identify two state-of-the-art methods for HAR with wearable sensor data. The first is an ensemble classifier, consisting of a decision tree, logistic regression, and a shallow multi-layer perceptron, proposed in 2015 by Catal et al. [12]. This ensemble achieved subject-dependent accuracy of 92% and 81% on MHEALTH and PAMAP2, respectively, with an average of 76% across all six datasets, and subject-independent accuracy of 95% and 85% on MHEALTH and PAMAP2, respectively, with an average of 69%. The second method that emerged as the state-of-the-art is the convolutional neural network (CNN) proposed in 2015 by Chen and Xue [13]. This CNN achieved subject-dependent accuracy of 90% and 82% on MHEALTH and PAMAP2, respectively, with an average of 83%, and subject-independent accuracy of 89% and 83% on MHEALTH and PAMAP2, respectively, with an average of 78%. Note that the 7–10 percentage points difference between the averages is largely due to the ensemble’s poor performance on the dataset to which the study’s authors could not apply the CNN. The authors’ analysis concludes that, although deep neural networks such as CNNs have achieved remarkable results in HAR with wearable sensor data, machine learning algorithms operating on handpicked features can in many cases achieve comparable results. In 2019, Abdu-Aguye and Gomaa [14] proposed an approach to feature extraction for sensor-based HAR. Their approach applies a wavelet transform and subjects the resulting decompositions to spatial pyramid pooling [15] to obtain fixed-length features which preserve both local and global patterns of the input signals. They used these features as inputs to a random forest, and compared the performance against a CNN. They estimated the subject-dependent performance by averaging over 15 repeated 75%/25% train/test splits. Their method achieved a subject-dependent accuracy of 89% and an F-score of 90%, while the CNN achieved an accuracy of 87% and an F-score of 87% on the REALWORLD dataset [16]. In 2020, Vakili et al. [17] evaluated seven machine learning algorithms, presumably operating on a set of extracted features, an artificial and a convolutional neural network, and a long short-term memory (LSTM) model on a range of internet-of-things datasets. The datasets’ application domains range from occupancy detection from environmental sensors, to rain prediction from weather station data, to HAR from wearable IMUs. They found that random forests outperformed the other methods on the SIMFALL [18] dataset with an average subject-dependent accuracy of 76%, estimated via 10-fold cross-validation. In 2018, Alharbi and Farrahi [19] proposed a CNN to recognise smoking activities from a smartphone and smartwatch’s inertial measurement units. They evaluated their approach on the UTSMOKE [20] dataset, but excluded the walking, sitting, and standing activities “because they were very simple to classify using the CNN.” They report an average subject-independent F1 score of 90%, estimated via a 70%/15%/15% train/validation/test split, but do not clarify how these splits can be construed in a subject-independent manner. This concludes our review of the state of the art in human activity recognition with respect to the datasets also used in this paper. We conclude our discussion of related work by reviewing the few papers that directly investigate the relationship between subject-dependent and subject-independent HAR performance.

Bao and Intille [21] assessed the subject-dependent performance of PSMs for recognising 20 activities of daily living (ADLs) across 20 users by training four learning algorithms on a set of semi-controlled laboratory data and evaluating them on a set of semi-naturalistic data, and the subject-independent performance of a PIM by performing a leave-one-subject-out cross-validation on the combined data from both sets. In a second experiment, they assess the subject-dependent performance of a PSM trained on three additional users’ laboratory data, and the subject-independent performance of a PIM trained on five different users’ laboratory data, using the three new users’ semi-naturalistic data for evaluation. Unfortunately, the differences in the protocols for estimating subject-dependent and subject-independent performance in the first experiment means that we cannot compare them directly (the latter accuracies are 18% to 50% *higher* than the former). Their second experiment, which affords a fairer comparison, directly contradicts these findings: the subject-dependent PSM accuracy (77%) exceeds the subject-independent PIM accuracy (73%) by 6%. Weiss and Lockhart [22] assessed the subject-independent and subject-dependent performances of PIMs, and the subject-dependent performances of PSMs for recognising six ADLs across 59 users and eight learning algorithms. They report that PSMs outperform a PIM by 2% to 30% on subject-dependent accuracy, and that a PIM achieves an 11% to 41% higher subject-dependent than subject-independent accuracy. These results suggest that, all else being equal, HAR methods will indeed perform better on data from known users than on data from unknown users. However, they tell us little about the size of the difference for a given personalisation–generalisation approach or about how the trade-off between subject-dependent and independent performance relates to the personalisation–generalisation approach.

## 3. Methods

We follow the standard approach to human activity recognition comprised of data pre-processing, segmentation into windows, feature extraction from those windows, and activity inference on them based on their features [23]—where the inference step is implemented with machine learning algorithms. We estimate and compare the performance of four popular machine learning algorithms—L_2_ (Ridge) regularised logistic regression, k-nearest neighbours (kNN), support vector machines (SVM), and a gradient boosted ensemble of decision trees (GBT)—using a set of features extracted from eight HAR datasets, which are summarised in Table 1.

For each dataset, Table 1 cites the relevant publication, and lists the number of activities (act) and people (ind), and the average number of trials per activity (±standard error) and sampling frequency (Hz). We chose datasets that were acquired via wearable inertial measurement units (IMU) comprised of an acceleration and angular velocity sensor, and worn either on the chest or the wrist. When sensors were to be worn on both wrists we chose the one associated with the right wrist. Unfortunately, the information about whether a user is right or left handed is unavailable for most datasets, making it impossible to choose the dominant wrist consistently. All datasets, except REALWORLD and SAFESENS which used a chest-worn sensor only, used a wrist-worn sensor, and only two datasets—PAMAP2 and SIMFALL—employed both a wrist and a chest-worn sensor. Figure 1 illustrates how the instances—each of which corresponds to the features extracted from one window—are distributed among the activities. Note that instead of distinguishing falls from ADLs in the SIMFALL dataset, which [18] were able to do with sensitivity, specificity, and accuracy all >99%, we focus on the 16 ADLs shown in the figure. Most of the activity labels are self-explanatory, but some of the activities in the UTSMOKE dataset merit further explanation. “SmokeST” denotes “Smoke Sitting”—smoking (presumably a cigarette) while sitting down—while “SmokeSD” denotes “Smoke Standing”—smoking while standing up. Similarly, “DrinkST” and “DrinkSD” denote “Drink Sitting” (drinking while sitting down) and “Drink Standing” (drinking while standing up), respectively.

The various cross-validation strategies, machine learning algorithms, and calculations of performance measures were implemented in Python (version 3.7.3), using the *scipy* ([25] version 1.1.0), *numpy* ([26] version 1.16.2), *pandas* ([27] version 0.23.3), and *sklearn* ([28] version 0.20.2) libraries, and parallelised via GNU parallel ([29] version 20161222). Analysis—t-tests, mixed effects models, and estimated marginal means—and all visualisations, were implemented in *R* ([30] version 3.6.1), wherein we used the mixed effects model implementation from the *lme4* library by ([31] version 1.1), and the estimated marginal means implementation from the *emmeans* library by ([32] version 1.3.2).

We propose and consider another personalisation–generalisation approach in addition to the person-independent model (PIM) and person-specific model (PSM), which we term an ensemble of PSMs (EPSM). An EPSM maintains a PSM for each known user. When an instance for a known user needs to be classified, an EPSM simply applies that user’s PSM, but when an instance originates with an unknown user, it applies each user’s PSM to obtain confidence scores (e.g., the estimated probability) for each activity of interest. Then the EPSM calculates each activity’s mean score, and classifies the instance to the activity with the maximum mean score. To deal with the (very few) users for whom the data do not cover all the activities of interest, and whose PSMs are therefore unaware of some activities and hence unable to generate a confidence score for those activities, we assume that those activities have a probability of zero. This is not unreasonable if we accept that some people will never perform certain activities (e.g., smoking, military crawling).

In addition to the basic, unweighted EPSM described above, this paper proposes two flavours of weighted EPSMs. A weighted EPSM makes predictions for known users in the same manner as the basic EPSM described above, but when making predictions for unknown users, a WEPSM combines its constituent models’ predictions via a weighted average. The two types of weighted EPSMs proposed in this paper differ in how the weights are determined. The first type is the κ-weighted EPSM (WEPSM). A WEPSM weights its constituents’ predictions according to each PSM’s average κ across all the other training users—i.e., all users except the one whose data are held out for testing and the one whose data (were) used for fitting the PSM. The second type is the baseline-feature-weighted EPSM (WEPSM_bf_). A baseline-feature-weighted EPSM weights its constituents’ predictions according to the mean euclidean distance between the PSM (training) user’s baseline features and the test user’s baseline features. A user’s baseline features are the features extracted from a random instance (window) of standing or sitting. To obtain the weight for a given training-user and test-user, we calculate the pairwise distances between the two users’ baseline features, and take the mean of these distances to weight the training user’s PSM predictions for the test user when aggregating them.

Figure 2 provides a graphical overview of the experiment we conducted, whose details are explained in the following subsections. The raw sensor signals from each dataset and sensor are first subjected to pre-processing. The pre-processed data—extracted features and labels, along with metadata identifying the originating dataset, sensor, and user—are then used to evaluate each learning algorithm’s ability to infer human activities in terms of subject-dependent (the figure’s left-hand branch) and subject-independent (the right-hand branch) performance. The figure also illustrates the difference between PSMs, unweighted EPSMs, κ-weighted EPSMs (WEPSM), and baseline-feature-weighted EPSMs (labelled "WEPSM_BF" in the figure).

### 3.1. Pre-Processing, Segmentation, and Feature Extraction

Some datasets come with a constant timestamp for each trial—presumably introduced when attempting to store POSIX^®^ epoch timestamps in (sub)millisecond resolution in Microsoft^®^ Excel^®^ spreadsheets. For these datasets we generate timestamps with a fixed inter-arrival time equal to the dataset’s nominal sampling frequency. Then, we (automatically) separate the raw data into non-overlapping natural trials by splitting the signal whenever the activity (label) changes or the inter-arrival time (i.e., the time between two subsequent samples) exceeds 1.5 s. To ensure that we have at least two trials per user and activity, each of the natural trials is then split into non-overlapping batches of 15 s. Next, the body and gravity components of each trial’s accelerometer signal are separated by the elliptical infinite-impulse response (IIR) low pass filter separates described by [33]. After discarding the original accelerometer data—which do not contain any information beyond that in the gravity and body components—we are left with three tri-axial signals: the gyroscope signal, the body acceleration signal, and the gravity acceleration signal. Finally, a set of time and frequency-domain features is extracted along a sliding 3 s window with 50% (1.5 s) overlap from each trial. From the angular velocity signal and both acceleration components we extract the mean, standard deviation, skew, and kurtosis, and from the angular velocity and body acceleration signal, the spectral power entropy, peak-power frequency, signal magnitude area, and the pairwise correlations between each signal’s axes. This amounts to a total of 84 features that are extracted from each window.

### 3.2. Activity Inference and Evaluation

We use logistic ridge regression with *C* = 0.98, a kNN classifier with *k* = 2 and weighted voting, a SVM classifier with a radial basis function with kernel coefficient γ = 0.001 and cost penalty *C* = 316, and a GBT with a learning rate α = 0.02 and comprised of 750 trees. The parameters for kNN, SVM, and GBT are taken from [24], who tuned them for subject-independent performance on the 17 activities in the SAFESENS dataset. The ridge parameter of *C* = 0.98 corresponds to weak regularisation, and was chosen to counteract the impact of correlated features. All features are standardised ($[x-\bar{x}]/s$) according to each feature’s mean ($\bar{x}$) and standard deviation (*s*) in the training data. We use Cohen’s kappa (κ) to quantify the predictive performance because—unlike other performance metrics such as sensitivity, specificity, and accuracy—it corrects for the probability of obtaining the observed level of agreement between the ground truth and predicted labels by chance, and because it is designed to measure predictive performance for multi-class classification.

To estimate an algorithm’s subject-dependent performance, the trials are used to generate the folds in a *k*-fold cross validation, a method we call leave-trials-out cross-validation [10]. Leave-trials-out cross-validation ensures that the raw data used to derive an instance in a training split are never used to derive the instances that constitute the corresponding test split, an issue that is bound to occur when working with instances derived from partially overlapping sliding windows [10], as we do here. PIM performance for known users is estimated by carrying out a *k*-fold leave-trials-out cross-validation across all the users in each dataset, and PSM performance by carrying out a separate *k*-fold leave-trials-out cross-validation for each user. In both cases k=n, where *n* denotes the number of people in the dataset. To estimate the subject-independent performance, we carry out a leave-*m*-users-out cross-validation with m=1 for EPSM and PIM, and m=n−1 for PSM.

## 4. Results and Analysis

Figure 3 illustrates the trade-off between the subject-dependent performance—i.e., the performance for users who were represented in the data used for training the model—on the horizontal axis, and the subject-independent performance—the performance for users who were not represented in the training data—on the vertical axis. In this figure, each datum corresponds to a single person (user), except in the case of person-specific models, where it corresponds to the median performance a model trained on data from the known user achieved on the other users in the dataset. The symbol and colour indicate which personalisation–generalisation approach (PIM, PSM, EPSM, WEPSM, or WEPSM${}_{\mathrm{bf}}$) was used. Table 2 summarises the results depicted in Figure 3, but using the PSM performance for all rather than, as shown in the figure, only that for the average unknown user. The table lists the mean κ (in %) ± standard error for each personalisation–generalisation approach, machine learning algorithm, dataset, and sensor location. To make the results more comparable with other results for these same datasets, the appendix presents the same tables with the accuracy (Table A1), error rate (Table A2), and weighted F-score (Table A3).

Inspecting these results, it is clear that the subject-independent performance is systematically and substantially worse than the subject-dependent performance. It is also clear that PSMs perform worse than PIMs in terms of their subject-independent performance. Furthermore, it is clear that a gradient boosted ensemble of decision trees (GBT) generally outperforms logistic regression (logreg) and k-nearest neighbours (kNN), with few exceptions. The most notable of these is the subject-independent performance with PSM, EPSM, WEPSM, and WEPSM${}_{\mathrm{bf}}$ on the SAFESENS dataset, where both kNN and logistic regression outperform gradient boosted trees, in some cases by more than one standard error. However, things are less clear when it comes to comparing PIMs and PSMs in terms of their subject-dependent performance, comparing PIMs and EPSMs in terms of their subject-independent performance, or comparing the different types of EPSMs against each other. To elucidate these matters, and to quantify the obvious differences mentioned above, we turn to statistical analyses which are discussed in the remainder of this section.

### 4.1. Analysis of the Subject-Dependent Performance

We can pair the performance when a person-specific model (PSM) is combined with a machine learning algorithm and applied to the data from a known person for a given dataset and sensor, to the performance when the same algorithm is combined with a person-independent model (PIM) and applied to the same dataset, sensor, and person. A paired t-test of these data yields a 95% confidence interval (C.I.) of 4.1 to 5.2 percentage points (hereafter, points) for the difference between the κ achieved with PSMs and that achieved with PIM, with a mean difference of 4.6 points ($t_{442}=16.2$, *p* < 2.2 × $10^{-16}$), suggesting that we can be 95% confident that a PSM outperforms a PIM on data from known users by 4.1 to 5.2 points. However, it is unlikely that the *t*-test’s underlying assumption of identically and independently distributed (IID) data is met, because the difference in the subject-dependent performance between PIM and PSM might depend not only on the dataset—which is expected due to the different activities of interest, and evident in Figure 3—but also on the learning algorithm.

A more appropriate tool for analysing data, such as these, which are not IID, is the linear mixed effects model (LMM). A LMM extends linear regression with so-called random effects which allow us to impose structure on the residuals. We can, for example, specify that the performances within datasets are correlated, or even that the difference in performance between personalisation–generalisation approaches varies depending on the dataset. The random effects are assumed to add up to zero, and hence the fixed effects (which are analogous to linear regression coefficients) can be estimated via (restricted) maximum likelihood. A LMM, like the linear regression model which it is based on, is built on the assumption of normally distributed residuals. The generalised linear mixed effects model (GLMM) extends the LMM to non-normal data, analogously to the generalised linear model (GLM). Like the GLM, it employs a link function, such as the logit, and error distribution (from the exponential family) to model non-normal data. We use logistic GLMMs—a GLMM with a logistic link function, and a binomial error distribution—to analyse the subject-dependent performance and subject-independent performance, and the difference between them. The explanatory variables considered as fixed effects were those of the personalisation–generalisation approach (PGA)—e.g., PIM, PSM, EPSM, and WEPSM; those of the machine learning algorithm (MLA); and the random effects of the dataset and sensor. For a detailed treatment of LMMs and GLMMs we refer interested readers to [34].

We use a GLMM to model the subject-dependent performance as a combination of (fixed) effects for the machine learning algorithm and personalisation–generalisation approach—either PIM or PSM, since subject-dependent EPSM performance is identical to that of its constituent PSMs—and a random effect to control for the variation of the personalisation–generalisation approach effect between datasets. This model explains the observed variation in the response with a residual standard deviation of 1.0157 between datasets, and with a standard deviation of 0.0255 between datasets’ sensors. This model reveals that the (random) effect of applying PSM varies with a standard deviation of 0.2929 between datasets, where it is very weakly negatively correlated (−0.11) with PIM performance, and with a standard deviation of 0.0407 between datasets’ sensors, where it is strongly positively correlated (0.75) with PIM performance. This confirms the intuition that a PSM likely confers less advantage when applied to a dataset on which a PIM performs well. The maximum likelihood estimates of the fixed effects, which are shown in Table 3, indicate that GBT—with an estimated κ of 89.4% and a 95% C.I. of 80.6% to 94.4% when used as a PIM—outperforms SVM by 19.5% (18.5% to 20.5%), logistic regression by 45.5% (44.9% to 46.2%), and kNN by 46% (45.3% to 46.6%), regardless of whether they are combined with a PIM or a PSM. They further show that PSMs outperform the corresponding PIM by 43.5% (17% to 76%, *p* = 0.00058) on subject-dependent performance.

Both the paired *t*-test and the GLMM agree that PSMs outperform PIMs in terms of subject-dependent performance. The GLMM estimates that PSMs outperform PIMs by 17% to 76% (with a mean of 43.5%) on subject-dependent performance in terms of the odds-ratio, the t-test estimates that difference as 4.1 to 5.2 points with a mean of 4.6 points. These estimates are consistent with each other. The GLMM estimate for GBT with PSM is κ = 92.3% with a C.I. of 85.5% to 96.1%. With PIM it is κ = 89.4% with a C.I. of 80.6% to 94.4%; 89.4% + 4.6 points equates to 94%, which is only 1.7 points above the GLMM estimate and well within the C.I. of 85.5% to 96.1% postulated by the GLMM. For kNN, the GLMM estimates the κ achieved with PSM at 86.7% with a C.I. of 76.2% to 93%, while the κ achieved with PIM is estimated at 82% with a C.I. of 69.2% to 90.2%; 82% + 4.6 equates to 86.6%, which not only lies well within the C.I. postulated by the GLMM, but is exceedingly close to the point estimate of 86.7%. It is similar for logistic regression: the GLMM estimates the κ achieved with PSM at 86.8% with a C.I. of 76.3% to 93.1%, and with PIM at 82.1% with a C.I. of 69.4% to 90.3%; 82.1% + 4.6 equates to 86.7%, which is only 0.1 point below the GLMM point estimate (and well withing the C.I. postulated by the GLMM). For SVM, the GLMM estimates the κ achieved with PSM at 90.7% with a C.I. of 82.6% to 95.2%, and the κ achieved with PIM at 87.1% with a C.I. of 77% to 93.2%; 87.1% + 4.6 equates to 91.7%, which is only one point above the GLMM point estimate, and well within the 95% C.I. postulated by the GLMM.

Both the t-test and GLMM rely on statistical assumptions. To compare the evaluated methods without relying on statistical assumptions, we rank the methods by their κ within each user, dataset, and sensor. Figure 4 illustrates the distribution of each method’s (learning algorithm + PSM or PIM) ranks across all datasets, sensors, and users. While no single method dominates across all users, the figure shows that person-specific models with gradient boosted ensembles of decision trees (GBT) perform better than all other methods for nearly 80% of users, and better than all but one method for over 10% of users, which confirms our statistical analysis. It is also clear that PSMs more often than not outperform a PIM for any given algorithm.

### 4.2. Analysis of the Subject-Independent Performance

A binomial logistic GLMM with fixed effects for the learning algorithm and personalisation–generalisation approach, and a random effect for sensors nested within datasets explains the variation in subject-independent performance with a residual error that varies with a standard deviation of 0.781 between datasets and with a standard deviation of 0.0218 between datasets’ sensors. The maximum likelihood estimates of the fixed effects, which are shown in Table 4, indicate that GBT—with an estimated κ of 71.3% (59.2% to 81.0%, *p* = 0.001) when used as a PIM—outperforms kNN by 20.2% (19.7% to 20.6%), logistic regression by 21.7% (21.3% to 22.2%), and SVM by 15.8% (15.3% to 16.3%). PIM outperforms PSM by 55.9% (55.3% to 56.2%), EPSM by 17.5% (17% to 18.1%), WEPSM by 16.4% (15.8% to 16.9%), and WEPSM${}_{\mathrm{bf}}$ by 18.4% (17.8% to 18.9%). All *p*-values < 2 × 10${}^{-16}$. This analysis clearly shows that PIM performs better for unknown users (i.e., on subject-independent performance) than the other personalisation–generalisation approaches, and that ensembles of PSMs perform better than a PSM. However, because the fixed effects for the different EPSMs—unweighted (EPSM), κ-weighted (WEPSM), or baseline-feature-weighted (WEPSM${}_{\mathrm{bf}}$)—estimate the difference between the particular EPSM and PIMs, and because their estimates are quite similar, this analysis on its own cannot compare the different types of EPSMs. To compare the different types of EPSMs we employ paired t-tests and estimated marginal means (also known as least-squares means) analysis.

According to the estimated marginal means, which are shown in Figure 5, the odds achieved by κ-weighted EPSMs are 1.4% higher than those achieved by unweighted EPSMs (*p* = 0.0009), which in turn are 1.1% higher than those achieved by EPSMs weighted by the inverse distance between the train and test user’s baseline features (*p* = 0.0159). A paired t-test of the difference in the subject-independent performance between WEPSM and EPSM yields a mean difference of 0.32 points with a 95% C.I. of 0.20 to 0.44 points, and a t-value of 17 on 443 degrees of freedom, which corresponds to a *p*-value of 3.48 × 10${}^{-7}$. This shows that a weighted EPSM significantly (albeit by less than one third of a point) outperforms an unweighted EPSM. A paired t-test of the difference in the subject-independent performance between EPSM and WEPSM${}_{\mathrm{bf}}$ yields a mean difference of 0.38 points, a 95% C.I. of 0.21 to 0.55 points, and a t-value of 4.45 on 435 degrees of freedom, corresponding to a *p*-value of 1.1 × 10${}^{-5}$. This shows that using baseline features for weighting the PSM predictions performs significantly worse (albeit by little more than one third of a point) than an unweighted EPSM. Both the estimated marginal means and paired t-tests lead to the conclusion that κ-weighted EPSMs significantly outperform unweighted EPSMs, which in turn significantly outperform EPSMs that are weighted by the inverse mean distance between the train and test user’s baseline features.

Analogously to the subject-dependent ranks shown in Figure 4, Figure 6 illustrates the distribution of each method’s (personalisation–generalisation approach + learning algorithm) subject-independent ranks across all datasets, sensors, and users. The figure clearly shows that PIM + GBT ranks first or second for over 60% of users—more often than any other method—and third or fourth for over 20% of users, which confirms the statistical analysis’s finding that PIM + GBT outperforms other methods on subject-independent performance. They also show that PIMs tend to perform better than other personalisation–generalisation approaches, in particular PSM, whose ranks are mostly in the bottom third. We can also see how a κ-weighted EPSM tends to shift the rather flat distribution of unweighted EPSMs slightly towards the left, towards the better (lower) ranks, particularly when combined with kNN or GBT.

### 4.3. Comparison of Subject-Dependent and Independent Performance

We use a binomial logistic GLMM with a fixed effect for the performance type (subject-dependent or subject-independent), one for the personalisation–generalisation approach (PIM, PSM, EPSM, etc.), and one for the interaction between them. There are two random effects, one for sensors nested within datasets, and one for the four learning algorithms. The reference level (intercept) corresponds to the subject-dependent performance achieved by PIMs. This model shows that the subject-dependent performance varies with a standard deviation of 0.1770 between learning algorithms, 0.825 between datasets, and 0.0382 between datasets’ sensors. The model’s estimates for the fixed effects are shown in Table 5, along with their 95% C.I.s and *p*-values. According to these estimates, PIMs achieve a subject-dependent κ, averaged over learning algorithms, of 82.2% (71.8% to 89.3%, *p* = 4.8 × 10${}^{-7}$) and PSMs outperform PIMs by 46% (44.9% to 47.2%, *p* < 2 × 10${}^{-16}$) in terms of the subject-dependent odds, with an estimated mean κ of 87.1% and a C.I. of 78.9% to 89.3% (according to the estimated marginal means shown in Figure 7). The subject-independent odds of PIMs are estimated at 48.1% (47.7% to 48.4%) of their subject-dependent odds, with a κ of 69% and an (estimated marginal means) C.I. of 55.1% to 80.1%. The subject-independent odds of PSMs are 13.6% of their subject-dependent odds, with a κ of 47.9% and an estimated marginal means C.I. of 33.7% to 62.4%. The subject-independent odds of EPSMs are 27% of their subject-dependent odds, with a κ of 64.6% and estimated marginal means C.I. of 50.2% to 76.8%. The subject-independent odds of WEPSMs are 27.4% of their subject-dependent odds, with a κ of 64.9% and an estimated marginal means C.I. of 50.6% to 77%. The subject-independent odds of WEPSM${}_{\mathrm{bf}}$ are 26.8% of their subject-dependent odds, with a κ of 64.4% and an estimated marginal means C.I. of 50% to 76.6%.

## 5. Discussion

Our analysis of the results shows that, on average, the best subject-dependent performance is achieved with PSMs and the best subject-independent performance with PIMs. Hence, in order to simultaneously optimise subject-dependent and subject-independent performance, we should use a PIM for unknown users and PSMs for known users wherever possible. If we use a PIM, rather than a PSM, to make predictions for known users we forego an expected improvement of over 43% in terms of the odds of a correct classification. For the datasets and models investigated in this paper, this corresponds to 4.1 to 5.6 percentage points difference in Cohen’s κ. If, on the other hand, we use a PSM rather than a PIM for unknown users we forego an expected improvement of nearly 56% in terms of the odds of a correct classification. If a PIM is not practicable—e.g., because we do not have access to the original training data when the time comes to integrate new users’ data into the HAR model—then an ensemble of PSMs can be employed. Among the three approaches for forming an EPSM which we considered in this paper, the κ-weighted EPSM emerged as a slightly, but significantly, better method for forming an EPSM than an unweighted EPSM, or a baseline-feature-weighted EPSM. Although a PIM performs significantly better than a κ-weighted EPSM (WEPSM), the difference in odds is estimated at a mere 16.4%, which is only 0.6 percentage points bigger than the difference in subject-independent κ between GBT and SVM, the two best-performing learning algorithms. Our analysis further shows that GBT significantly outperforms kNN, L${}_{2}$-regularised logistic regression, and SVM, but the latter by only 15.2%.

The state of the art for the FUSION dataset achieves a subject-dependent accuracy of > 99% [4]. Our GBT PIM achieves an accuracy of 98.3% ± 0.3 and our GBT PSM an accuracy of 98% ± 0.3 on this dataset. On the OPPORT dataset, the state of the art achieves a subject-dependent error rate of 18% [5] and F-score of 93% [6,7]. Our GBT PSM achieves a subject-dependent error rate of 10.9% ± 1.5 (GBT PIM: 12.2% ± 1.9) and F-score of 89.1% ± 1.5 (GBT PIM: 87.8% ± 1.9) on this dataset. According to Jordao et al. [10], the state of the art for the MHEALTH dataset achieves a subject-dependent accuracy of 92% (with a C.I. of 88% to 96%), and a subject-independent accuracy of 95% (C.I.: 91% to 98%). Our GBT PIM achieves a subject-dependent accuracy of 97.8% ± 0.7 (GBT PSM: 97.5% ± 1.1), and a subject-independent accuracy of 84% ± 3.3 on this dataset. Following the same paper [10], the state of the art for the PAMAP2 dataset achieves a subject-dependent accuracy of 82% (C.I.: 77% to 88%) and a subject-independent accuracy of 85% (C.I.: 76% to 94%). Our GBT PSM achieves a subject-dependent accuracy of 88.9% ± 0.6 (GBT PIM: 88.8% ± 0.4) when using the chest-mounted sensor, and our GBT PIM a subject-independent accuracy of 80.6% ± 2.6 when using the wrist-mounted sensor from this dataset. On the REALWORLD dataset, the state of the art achieves a subject-dependent accuracy of 89% and F-score of 90% [16]. Our GBT PSM achieves a subject-dependent accuracy of 96.9% ± 0.3 (GBT PIM: 94.5% ± 0.5) and F-score of 96.8% ± 0.5 (GBT PIM: 94.7% ± 0.5) on this dataset. The state of the art for the SIMFALL dataset achieves a subject-dependent accuracy of 76% [17]. Our GBT PSM achieves a subject-dependent accuracy of 68.1% ± 1.2 (GBT PIM: 60% ± 1.1) when using the chest-mounted sensor and 65.1% ± 1.4 (GBT PIM: 58.3% ± 1.4) when using the wrist-mounted sensor from this dataset. On the UTSMOKE dataset, the state of the art achieves a subject-independent accuracy of 90% [19]. Our GBT PIM achieves a subject-independent accuracy of 73.2% ± 2.5 (SVM PIM: 73.6% ± 2.3) on this dataset. To summarise, our approach performs comparable (i.e., within the margin of error) to the state of the art for all but one (UTSMOKE) dataset in terms of the subject-independent performance. In terms of the subject-dependent performance, our approach performs clearly worse than the state of the art (by about 8 points) for one dataset (SIMFALL), within one percentage point for another (FUSION), and better than the state of the art for four datasets—namely the REALWORLD, UTSMOKE, MHEALTH, and PAMAP2 datasets. This shows that our gradient boosted ensemble of decision trees, combined with PSMs for subject-dependent or a PIM for subject-independent performance, performs comparably to the state-of-the-art for a wide range of HAR problems, without problem-specific feature engineering or tuning of model hyper-parameters.

## 6. Conclusions

This paper compared the subject-dependent and subject-independent performances of person-independent models (PIMs), person-specific models (PSMs), and three types of ensembles of PSMs (EPSMs)—unweighted, κ-weighted, and baseline-feature-weighted—when combined with four popular HAR algorithms across eight publicly available HAR datasets. An analysis with generalised linear mixed effects models (GLMM) showed that GBT significantly outperforms the other algorithms on both subject-dependent and subject-independent performance; that PSMs outperform a PIM by 43.5% (in terms of the odds of correct versus incorrect classification) in subject-dependent performance; and that a PIM outperforms PSMs by 55.9% and κ-weighted EPSM by 16.4% on subject-independent performance. Furthermore, our analysis of the subject-independent performance shows that κ-weighted EPSMs significantly outperform unweighted EPSMs, albeit by as little as 0.32 percentage points and 1.4% in terms of the odds, and that unweighted EPSMs in turn significantly outperform baseline-feature-weighted EPSMs by about the same amount.

Our approach—a gradient boosted ensemble of decision trees combined with person-specific models for known users and with a PIM for unknown users—performs comparably to the state of the art on one dataset and outperforms the state of the art on four datasets in terms of subject-dependent performance. In terms of the subject-independent performance, our approach performs comparably to the state of the art on all but one of the datasets, for which published results that include the subject-independent performance exist.

## Figures and Tables

**Figure 1 sensors-20-03647-f001:**
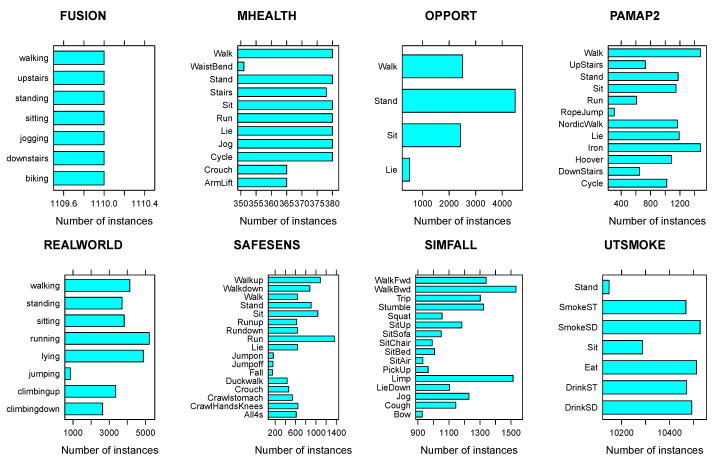
Number of instances per activity for each dataset.

**Figure 2 sensors-20-03647-f002:**
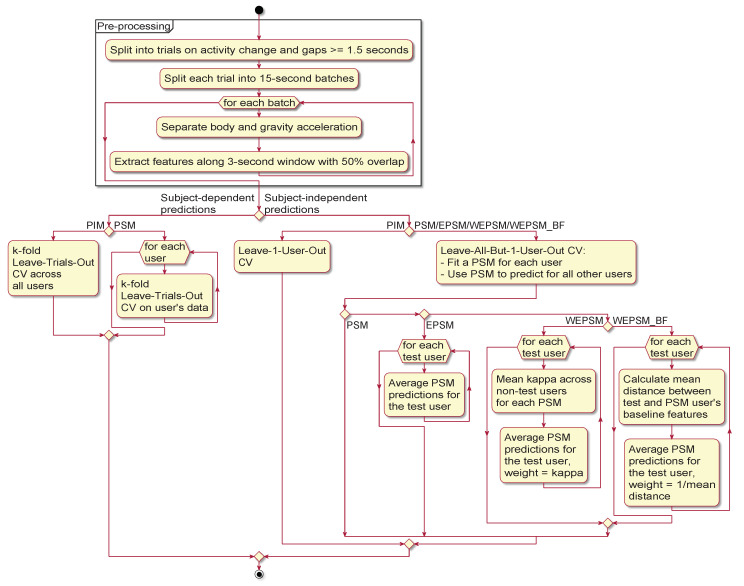
Graphical summary of the experiment. Each dataset is pre-processed once, and then used to obtain the subject-dependent and subject-independent predictions with each learning algorithm and personalisation–generalisation approach (person-independent models (PIMs), person-specific models (PSMs), etc.). Note that an ensemble of PSMs is identical to a PSM in the case of subject-dependent predictions (i.e., for known users).

**Figure 3 sensors-20-03647-f003:**
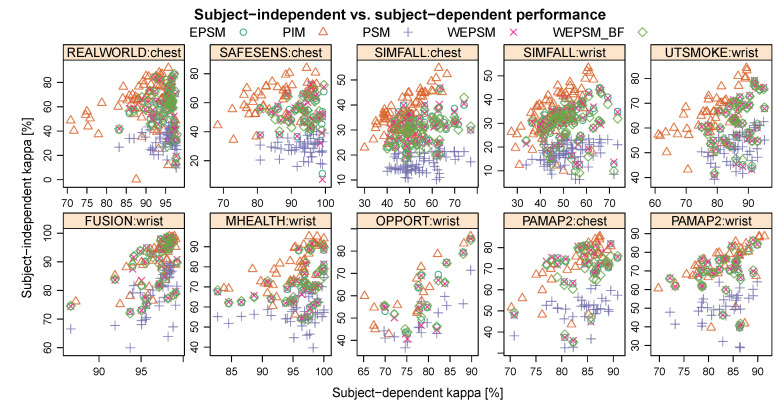
Subject-independent (vertical axis) versus subject-dependent (horizontal axis) κ (%) across all learning algorithms. Axes have been scaled to encompass the data for improved visibility. Note how the subject-dependent performance of (E)PSMs tends to be better (further to the right) than that of PIMs, and there is a clear difference in subject-independent performance between PIMs and PSMs.

**Figure 4 sensors-20-03647-f004:**
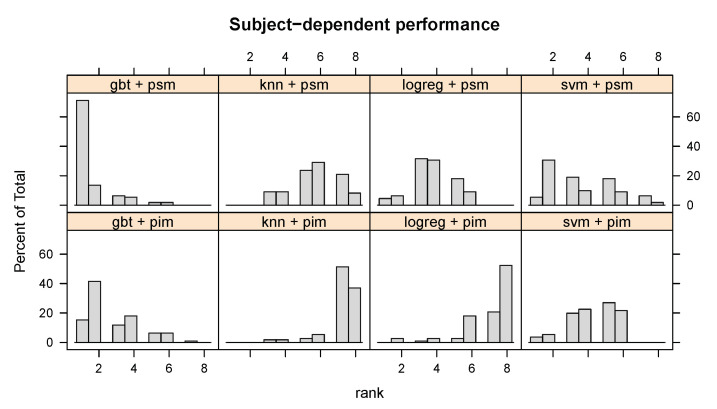
Distribution of the user-wise subject-dependent ranks of each learning algorithm + personalisation–generalisation approach. A person-specific gradient boosted ensemble of decision trees (GBT + PSM) outperforms the other methods for over 70% of users.

**Figure 5 sensors-20-03647-f005:**
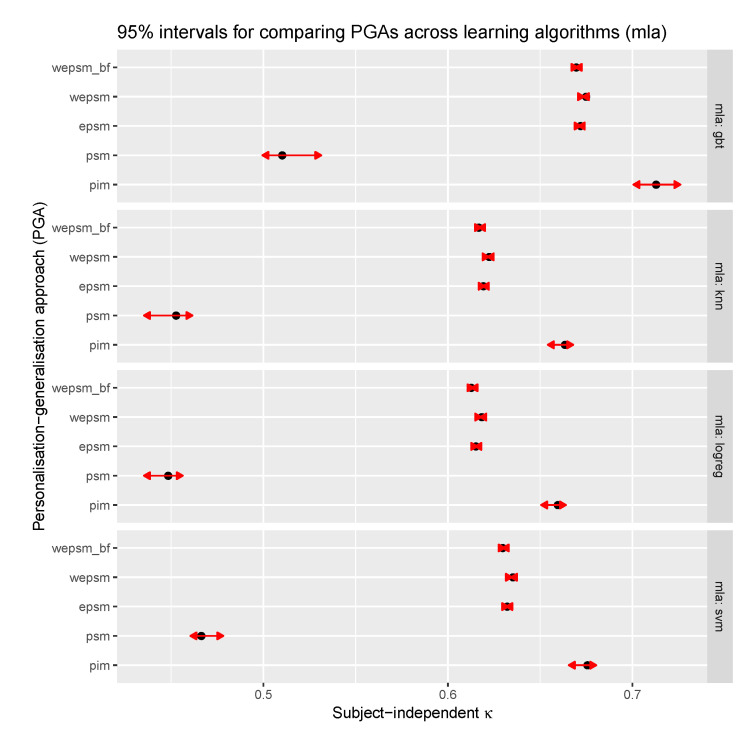
Estimated marginal means for comparing the subject-independent performance of personalisation–generalisation approaches across learning algorithms. Person-independent models (PIM) clearly outperform the other personalisation–generalisation approaches for any learning algorithm, and ensembles of person-specific models (particularly WEPSMs) clearly outperform its constituent person-specific models (PSM).

**Figure 6 sensors-20-03647-f006:**
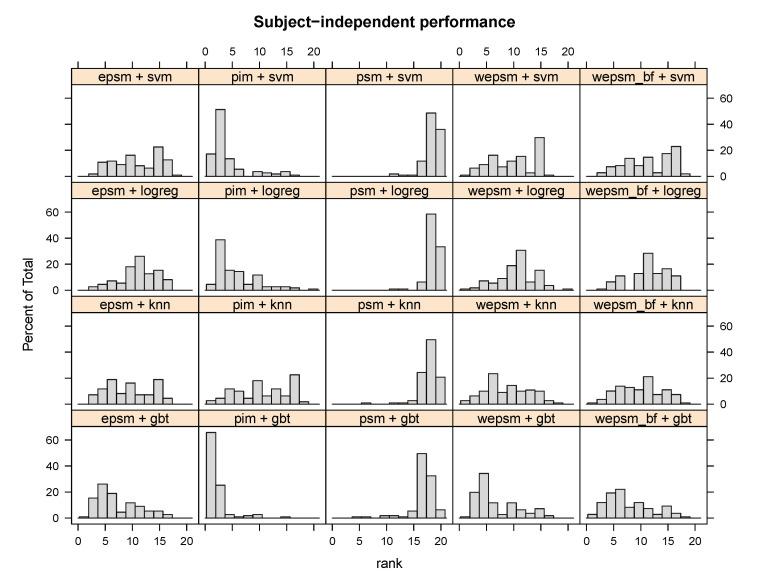
Distribution of the user-wise subject-independent ranks of each personalisation–generalisation approach + learning algorithm. PIM + GBT outperforms the other methods for over 60% of users, and performs second-best for over 20% of users.

**Figure 7 sensors-20-03647-f007:**
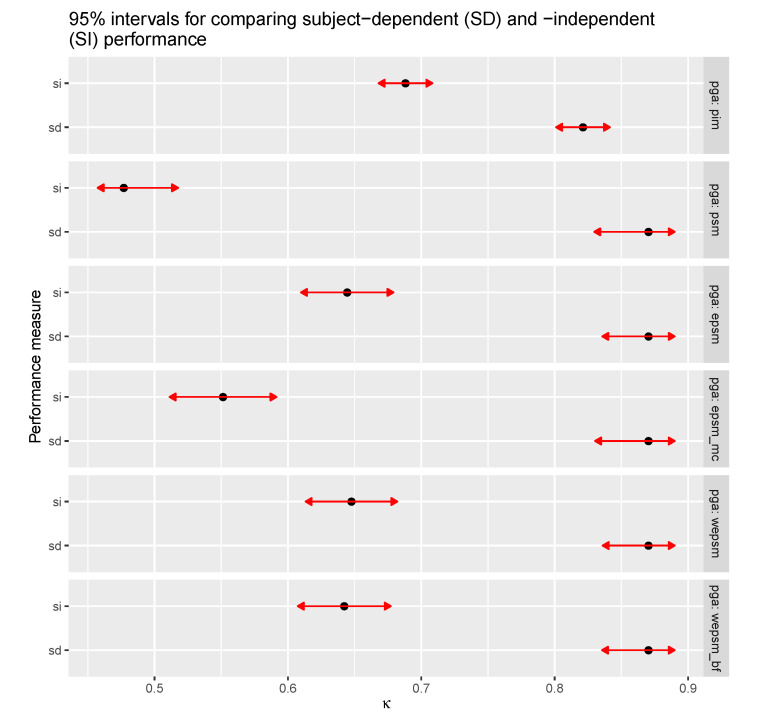
Estimated marginal means for the (average) difference between subject-dependent (SD) and subject-independent (SI) performance across personalisation–generalisation approaches. Subject-dependent performance is clearly better than subject-independent performance, a discrepancy that is minimised with person-independent models (PIM). The confidence intervals for the subject-independent performances of PIMs and ensembles of PSMs, particularly WEPSMs, overlap.

**Table 1 sensors-20-03647-t001:** Number of (act)ivities and (ind)ividuals, trials/activity (±standard error), and sampling frequency (Hz) for each of the datasets.

	Dataset	Act	Ind	Trials/Act	Hz
[8]	FUSION	7	10	90 ± 0	50
[11]	MHEALTH	11	10	38 ± 0	50
[3]	OPPORT	4	4	590 ± 258	30
[9]	PAMAP2	12	9	81 ± 8	100
[16]	REALWORLD	8	15	318 ± 42	50
[24]	SAFESENS	17	11	91 ± 13	33
[18]	SIMFALL	16	17	128 ± 8	25
[20]	UTSMOKE	7	11	859 ± 7	50

**Table 2 sensors-20-03647-t002:** Subject-dependent and subject-independent κ (%) ± standard error when learning algorithms (mla) are combined with a person-independent model (PIM), a person-specific model (PSM), an unweighted ensemble of PSMs (EPSM), a κ-weighted EPSM (WEPSM), or a baseline-feature-weighted EPSM (WEPSM${}_{\mathrm{bf}}$).

			Subject-Dependent		Subject-Independent				
Dataset	Sensor	mla	PIM	(E)PSM	PIM	PSM	EPSM	WEPSM	WEPSM${}_{\mathrm{bf}}$
FUSION	wrist	gbt	97.9 ± 0.3	97.6 ± 0.4	92.4 ± 2.2	81.4 ± 2.1	90.6 ± 2.6	90.7 ± 2.5	90.2 ± 2.6
		knn	94.0 ± 0.9	94.2 ± 1.0	85.9 ± 2.0	74.9 ± 2.6	87.5 ± 2.8	87.5 ± 2.8	87.3 ± 2.8
		logreg	96.7 ± 0.6	97.4 ± 0.4	91.9 ± 2.1	79.4 ± 2.7	89.5 ± 2.8	89.5 ± 2.8	89.0 ± 2.7
		svm	98.0 ± 0.3	97.8 ± 0.4	90.9 ± 2.1	80.0 ± 2.3	90.3 ± 2.7	90.4 ± 2.7	90.0 ± 2.6
MHEALTH	wrist	gbt	97.5 ± 0.8	97.2 ± 1.2	82.4 ± 3.6	59.5 ± 2.3	72.2 ± 3.5	72.4 ± 3.4	71.5 ± 3.5
		knn	92.9 ± 1.3	93.7 ± 1.4	76.1 ± 3.1	56.0 ± 2.1	71.4 ± 2.6	71.6 ± 2.6	72.8 ± 2.5
		logreg	93.2 ± 1.4	95.8 ± 1.4	78.9 ± 3.3	54.1 ± 2.4	70.0 ± 2.9	70.2 ± 2.9	70.8 ± 2.9
		svm	95.5 ± 1.0	96.8 ± 1.0	82.0 ± 2.6	58.0 ± 2.1	72.0 ± 3.9	72.1 ± 3.9	71.4 ± 4.0
OPPORT	wrist	gbt	81.5 ± 2.8	83.5 ± 2.4	69.0 ± 7.2	57.9 ± 5.8	66.5 ± 8.1	66.2 ± 8.4	66.7 ± 7.9
		knn	71.1 ± 2.7	74.8 ± 2.7	51.4 ± 4.2	42.9 ± 3.3	54.5 ± 5.4	53.6 ± 5.1	54.8 ± 4.6
		logreg	71.9 ± 3.7	76.7 ± 3.0	59.9 ± 7.0	46.2 ± 3.4	56.7 ± 6.7	56.4 ± 7.0	57.2 ± 6.4
		svm	80.3 ± 2.6	81.0 ± 2.4	65.4 ± 6.7	48.4 ± 2.8	61.7 ± 6.6	61.3 ± 7.0	62.3 ± 6.2
PAMAP2	chest	gbt	87.5 ± 0.5	87.7 ± 0.6	77.4 ± 4.3	54.7 ± 2.9	72.4 ± 4.1	72.9 ± 4.2	72.0 ± 3.8
		knn	75.5 ± 1.0	78.4 ± 1.2	63.7 ± 2.5	49.0 ± 1.8	67.7 ± 3.2	68.2 ± 3.4	66.3 ± 3.1
		logreg	82.5 ± 1.0	85.4 ± 0.9	72.2 ± 3.8	48.8 ± 2.4	69.4 ± 4.9	69.4 ± 4.9	68.6 ± 4.8
		svm	86.0 ± 0.7	85.1 ± 0.8	73.7 ± 4.5	49.7 ± 2.6	69.4 ± 5.1	70.0 ± 5.1	68.5 ± 5.1
PAMAP2	wrist	gbt	86.8 ± 1.1	86.0 ± 0.9	78.5 ± 2.8	56.8 ± 2.3	71.7 ± 2.7	72.2 ± 2.6	72.0 ± 2.8
		knn	77.4 ± 1.5	78.9 ± 1.6	65.2 ± 4.1	47.5 ± 2.7	68.1 ± 3.9	68.5 ± 4.0	67.6 ± 3.8
		logreg	82.4 ± 1.7	83.5 ± 1.3	74.7 ± 4.1	49.9 ± 3.5	68.8 ± 4.9	69.3 ± 4.9	69.0 ± 4.8
		svm	84.9 ± 1.3	83.3 ± 1.3	73.1 ± 5.1	46.6 ± 3.0	68.3 ± 4.5	68.8 ± 4.5	68.2 ± 4.3
REALWORLD	chest	gbt	93.3 ± 0.6	96.1 ± 0.4	71.7 ± 4.4	37.5 ± 1.9	62.7 ± 4.0	64.1 ± 3.8	63.2 ± 3.9
		knn	85.3 ± 1.5	91.3 ± 1.0	59.3 ± 3.4	37.9 ± 2.4	61.8 ± 3.7	62.8 ± 3.6	62.0 ± 3.8
		logreg	83.8 ± 1.8	95.4 ± 0.5	60.6 ± 5.8	30.3 ± 2.1	57.1 ± 4.2	58.7 ± 4.2	57.1 ± 4.3
		svm	92.0 ± 0.7	95.5 ± 0.4	62.2 ± 5.1	30.4 ± 2.1	54.8 ± 4.5	56.5 ± 4.3	55.0 ± 4.7
SAFESENS	chest	gbt	93.9 ± 0.9	97.0 ± 0.8	67.6 ± 3.3	27.9 ± 2.0	48.9 ± 4.8	48.7 ± 5.4	53.9 ± 3.3
		knn	81.3 ± 1.9	87.8 ± 1.5	55.7 ± 3.5	30.2 ± 1.7	54.7 ± 3.2	54.6 ± 3.3	54.2 ± 2.7
		logreg	78.7 ± 1.6	93.1 ± 1.0	64.1 ± 3.0	27.4 ± 1.8	54.0 ± 2.7	53.3 ± 2.7	54.1 ± 2.8
		svm	88.1 ± 1.2	95.2 ± 0.8	66.9 ± 2.7	29.9 ± 1.8	51.9 ± 2.6	53.0 ± 2.4	51.9 ± 2.9
SIMFALL	chest	gbt	57.2 ± 1.2	65.9 ± 1.3	43.9 ± 1.6	19.3 ± 0.7	33.5 ± 1.6	33.5 ± 1.6	32.6 ± 1.5
		knn	45.0 ± 0.9	49.5 ± 1.2	30.3 ± 0.7	19.8 ± 0.6	33.3 ± 1.0	33.2 ± 1.1	32.2 ± 0.9
		logreg	38.3 ± 0.9	52.3 ± 1.2	34.5 ± 1.1	14.5 ± 0.5	29.1 ± 1.0	29.1 ± 1.0	28.2 ± 1.0
		svm	50.1 ± 0.7	49.5 ± 1.6	38.2 ± 1.4	14.1 ± 0.4	25.9 ± 1.0	26.2 ± 0.9	24.8 ± 1.0
SIMFALL	wrist	gbt	55.4 ± 1.5	62.7 ± 1.5	40.8 ± 2.3	19.1 ± 1.0	32.9 ± 2.2	33.0 ± 2.2	32.0 ± 2.2
		knn	44.6 ± 1.2	48.8 ± 1.2	29.2 ± 1.5	19.2 ± 1.0	31.8 ± 1.6	31.9 ± 1.6	31.0 ± 1.6
		logreg	37.3 ± 1.4	49.6 ± 1.3	32.7 ± 2.1	16.2 ± 0.9	27.9 ± 1.7	28.4 ± 1.7	27.6 ± 1.8
		svm	48.2 ± 1.3	45.9 ± 1.5	36.0 ± 2.3	13.7 ± 0.7	27.1 ± 1.5	27.3 ± 1.6	26.5 ± 1.7
UTSMOKE	wrist	gbt	80.9 ± 1.5	90.8 ± 0.9	68.7 ± 2.9	54.8 ± 1.8	65.4 ± 3.2	65.4 ± 3.3	65.3 ± 3.1
		knn	76.3 ± 1.3	81.2 ± 1.2	61.6 ± 2.4	50.8 ± 1.7	60.7 ± 2.8	60.8 ± 2.9	60.5 ± 2.7
		logreg	68.9 ± 2.1	84.1 ± 1.2	63.2 ± 2.5	50.5 ± 1.6	59.4 ± 2.5	59.4 ± 2.5	59.6 ± 2.4
		svm	83.6 ± 1.3	89.1 ± 0.9	69.2 ± 2.7	52.9 ± 1.8	63.6 ± 2.9	63.8 ± 3.0	63.8 ± 2.7

**Table 3 sensors-20-03647-t003:** GLMM (Generalised Linear Mixed Effects Model) estimates (β), 95% confidence intervals, and *p*-values of the fixed effects on subject-dependent performance associated with learning algorithms and personalisation–generalisation approaches.

Coefficient	2.5%	*β*	97.5%	*p*
(Intercept)	1.425	2.129	2.833	3.1 × 10^−9^
kNN	−0.628	−0.616	−0.604	<2.0 × 10^−16^
logreg	−0.620	−0.608	−0.596	<2.0 × 10^−16^
SVM	−0.229	−0.217	−0.204	<2.0 × 10^−16^
PSM	0.155	0.361	0.566	5.78 × 10^−4^

**Table 4 sensors-20-03647-t004:** GLMM (Generalised Linear Mixed Effects Model) estimates (β), 95% confidence intervals, and *p*-values of the fixed effects on subject-independent performance associated with learning algorithms and personalisation–generalisation approaches.

Coefficient	2.5%	*β*	97.5%	*p*
(Intercept)	0.367	0.907	1.448	1.00 × 10^−3^
kNN	−0.231	−0.225	−0.219	<2.0 × 10^−16^
logreg	−0.251	−0.245	−0.239	<2.0 × 10^−16^
SVM	−0.178	−0.172	−0.166	<2.0 × 10^−16^
PSM	−0.825	−0.818	−0.812	<2.0 × 10^−16^
EPSM	−0.199	−0.193	−0.186	<2.0 × 10^−16^
WEPSM	−0.186	−0.179	−0.172	<2.0 × 10^−16^
WEPSM${}_{\mathrm{bf}}$	−0.210	−0.203	−0.196	<2.0 × 10^−16^

**Table 5 sensors-20-03647-t005:** GLMM (Generalised Linear Mixed Effects Model) estimates (β), 95% confidence intervals, and *p*-values of the fixed effects associated with subject-independent (SI) performance and personalisation–generalisation approaches.

Coefficient	2.5%	*β*	97.5%	*p*
(Intercept)	0.930	1.530	2.130	5.83 × 10^−7^
PSM/(E)PSM${}_{\left(\mathrm{bf}\right)}$	0.371	0.379	0.387	<2.0 × 10^−16^
SI	−0.739	−0.732	−0.725	<2.0 × 10^−16^
SI + EPSM	−0.586	−0.575	−0.564	<2.0 × 10^−16^
SI + PSM	−1.272	−1.261	−1.250	<2.0 × 10^−16^
SI + WEPSM	−0.572	−0.561	−0.551	<2.0 × 10^−16^
SI + WEPSM${}_{\mathrm{bf}}$	−0.595	−0.584	−0.574	<2.0 × 10^−16^

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
