# Peer review of "Comparing Person-Specific and Independent Models on Subject-Dependent and Independent Human Activity Recognition Performance"

_sensors, 2020, doi:10.3390/s20133647_

Round 1

Reviewer 1 Report

This article presents an interesting analysis of person-specific and independent models on subject-dependent and subject-independent human activity recognition performance.

The article is well written and tried to address an important issue in human activity recognition.

Besides,  the analysis done by the authors is quite an interesting one, but the authors need to present their results more comprehensively.

They address these issues pointed out below.

 # The presentation of results in sections 4.1 to 4.3 can be better represented in charts or graphs.

# The references are quite old, the references should be updated to reflect state-of-the-art.

Author Response

Point 1
=======

The presentation of results in sections 4.1 to 4.3 can be better represented in charts or graphs.

Response 1
==========

We have added three tables (Tables 3, 4, and 5 on page 11, 13, and 15, respectively) with the fixed-effects estimates, 95% confidence intervals, and P-values of the generalised linear mixed effects models. We have also added figures (Figures 5 and 7 on page 13 and 15, respectively) with graphical representations of the estimated marginal means and their 95% confidence intervals.

Point 2
=======

The references are quite old, the references should be updated to reflect state-of-the-art.

Response 2
==========

We identify the state-of-the-art for 4 out of the 8 data-sets from the 2019 survey by Wang et al. [2] or the 2019 paper by Jordao et al. [10] which compares several state-of-the-art HAR methods across multiple benchmark data-sets. For the remaining data-sets, we cite a paper published in 2019 by Abdu-Aguye and Gomaa [14] for the REALWORLD data-set, a paper published in 2020 by Vakili et al. [17] for the SIMFALL data-set, and a paper published in 2018 by Alharbi and Farrahi [19] for the UTSMOKE data-set. Our references' median publication year is 2015 (i.e., at least half of our sources are no more than five years old). Many of our older sources are either references to a paper introducing one of our eight benchmark data-sets, or software libraries. Unfortunately, we are not aware of more recent papers that address the topic of subject-dependent and -independent performance in HAR, or the use of ensemble classifiers for HAR.

Reviewer 2 Report

In this manuscript, the authors analyze the subject-dependent and -independent performance of person-independent models (PIM), person-specific model (PSM), and three types of ensembles of PSMs. They compare four machine learning algorithms using eight data sets. According to their results, PSMs outperform PIMs in terms of subject-dependent performance, and PIMs outperforms PSMs and ensembles of PSMs in terms of subject-independent performance. Here are my comments and questions:

  1. Could you please explain to the reader how you apply the particular classifiers to the dynamic data of various lengths? Please correct me if I am wrong. As far as I understood, you segment the data into windows, calculate the features, and perform the inference for each window. So, we obtain a classifier output for each window. How is the recognized activity estimated from a sequence of classifier responses?
  2. The concept of the weighted ensemble of PSMs is interesting. Could you explain more precisely how do you determine the weights in both your variants? The description given in lines 229 - 234 is a bit intricate.
  3. How do you separate the raw data into trials (lines 239-240)? Is it done automatically?

Author Response

Point 1
=======

Could you please explain to the reader how you apply the particular classifiers to the dynamic data of various lengths? Please correct me if I am wrong. As far as I understood, you segment the data into windows, calculate the features, and perform the inference for each window. So, we obtain a classifier output for each window. How is the recognized activity estimated from a sequence of classifier responses?

Response 1
==========

You are correct in that we obtain a classifier output for each window. We cannot conceive of a situation in which we would want to estimate a single activity for a (theoretically infinite) time series of sensor data.

We use a window size of 3 seconds, with an overlap of 50% (1.5 seconds). This means that our system produces its first output after 3 seconds, which is followed by the next output after an additional 1.5 seconds elapsed, and another after an additional 1.5 seconds, etc. Each output predicts the activity that the user was engaged in during the 3 seconds summarised by the window's features. If we need predictions at a lower time resolution (say, every minute or every 15 minutes), we can either extract the features along a larger window, or aggregate the predictions along a larger window. Which approach performs better for a given time resolution depends on the application, because human activities take place at such different time scales (consider, for example "taking a step" and "going for a run").

Point 2
=======

The concept of the weighted ensemble of PSMs is interesting. Could you explain more precisely how do you determine the weights in both your variants? The description given in lines 229 - 234 is a bit intricate.

Response 2
==========

We have added a graphical overview of the experiments (Figure 2 on page 7) and an accompanying paragraph (lines 240--247), which we hope clarifies this point.

Point 3
=======

How do you separate the raw data into trials (lines 239-240)? Is it done automatically?

Response 3
==========

The graphical summary of the experiment in Figure 2 and its accompanying paragraph, should clarify this point. We have also adapted the text you mention in your question (lines 252--254 in the revised manuscript) to clarify this.

Reviewer 3 Report

The paper's topic falls perfectly within the scope of the journal.

The paper builds on a previous paper by the same authors, as they themselves point out. However, the previous version has been enriched through the inclusion of new variants of a model (weighted EPSM) and a much more detailed statistical analysis.

The conclusions are quite as expected (PSM performing better than PIM on a specific user, and vice versa when we consider a set of unknown users), but a quantitative indication of the difference in performance is provided, which makes the comparison better grounded.

The statistical apparatus is quite sound.

I suggest the authors make the following changes/additions to their paper:

  • include a taxonomy of the experiments, since the average reader may get a bit lost into the overall combination of ML algorithms, models, training approaches, and datasets;
  • how do we read the algorithm used in figure 2?
  • the experiments are said to be performed in Python , while the analysis is said to be carried out in R. Please explain more clearly what was carried out in either language. Also, please include the fulls et of libraries employed in either language.
  • add a comparison of the algorithms free of statistical assumptions (whose violations could make the results of both t-test and LMM less reliable), e.g. through identifying the Pareto-dominant method/algorithm combination for each user datapoint (if any), i.e. the method\algorithm that has the better couple of kappa-values (not exceeded by another), and see how many times each method/algorithm is Pareto-dominant

Author Response

Point 1
=======

include a taxonomy of the experiments, since the average reader may get a bit lost into the overall combination of ML algorithms, models, training approaches, and datasets;

Response 1
==========

The revised manuscript now includes a graphical overview/summary of the experiment (Figure 2, page 7).

Point 2
=======

how do we read the algorithm used in figure 2?

Response 2
==========

Figure 2 (Figure 3 in the revised manuscript) shows the results across all algorithms, to highlight the differences between personalisation-generalisation approaches (PIM, PSM, ...). The detailed per-algorithm results can be found in Table 2, and Tables A1--A3.

Point 3
=======

the experiments are said to be performed in Python , while the analysis is said to be carried out in R. Please explain more clearly what was carried out in either language. Also, please include the fulls et of libraries employed in either language.

Response 3
==========

We have expanded the paragraph of the programming languages and environments we used, and what each one was used for (lines 211--216).

Point 4
=======

add a comparison of the algorithms free of statistical assumptions (whose violations could make the results of both t-test and LMM less reliable), e.g. through identifying the Pareto-dominant method/algorithm combination for each user datapoint (if any), i.e. the method/algorithm that has the better couple of kappa-values (not exceeded by another), and see how many times each method/algorithm is Pareto-dominant.

Response 4
==========

We have added two figures (Figures 4 and 6 on page 12 and 14, respectively), and accompanying explanatory paragraphs (lines 375--382, and lines 416--425), with histograms of each method's (algorithm + personalisation-generalisation approach) ranks, ranked by their kappa score within each data-set, sensor, and user.